# The genome of an intranuclear parasite, *Paramicrosporidium saccamoebae*, reveals alternative adaptations to obligate intracellular parasitism

C Alisha Quandt[1], Denis Beaudet[2], Daniele Corsaro[3], Julia Walochnik[4], Rolf Michel[5], Nicolas Corradi[2]*, Timothy Y James[1]*

[1]Department of Ecology and Evolutionary Biology, University of Michigan, Ann Arbor, United States; [2]Department of Biology, University of Ottawa, Ottawa, Canada; [3]CHLAREAS Chlamydia Research Association, Nancy, France; [4]Molecular Parasitology, Institute for Specific Prophylaxis and Tropical Medicine, Medical University of Vienna, Koblenz, Germany; [5]Laboratory of Electron Microscopy, Central Institute of the Federal Armed Forces Medical Services, Koblenz, Germany

**Abstract** Intracellular parasitism often results in gene loss, genome reduction, and dependence upon the host for cellular functioning. Rozellomycota is a clade comprising many such parasites and is related to the diverse, highly reduced, animal parasites, Microsporidia. We sequenced the nuclear and mitochondrial genomes of *Paramicrosporidium saccamoebae* [Rozellomycota], an intranuclear parasite of amoebae. A canonical fungal mitochondrial genome was recovered from *P. saccamoebae* that encodes genes necessary for the complete oxidative phosphorylation pathway including Complex I, differentiating it from most endoparasites including its sequenced relatives in Rozellomycota and Microsporidia. Comparative analysis revealed that *P. saccamoebae* shares more gene content with distantly related Fungi than with its closest relatives, suggesting that genome evolution in Rozellomycota and Microsporidia has been affected by repeated and independent gene losses, possibly as a result of variation in parasitic strategies (e.g. host and subcellular localization) or due to multiple transitions to parasitism.
DOI: https://doi.org/10.7554/eLife.29594.001

*For correspondence:
ncorradi@uottawa.ca (NC);
tyjames@umich.edu (TYJ)

**Competing interests:** The authors declare that no competing interests exist.

## Introduction

Near the phylogenetic root of the Kingdom Fungi lies a branch that unites Microsporidia and a group variably known as Rozellomycota or Cryptomycota. This clade of intracellular parasites are considered energy thieves, because they import ATP from their hosts. The clade encompasses a great diversity of morphologies from the fungus-like *Rozella* spp. to the highly derived Microsporidia which are best known for the reductive nature of their cells (*Hirt et al., 1997*; *Williams et al., 2002*; *Burri et al., 2006*). The biology of Rozellomycota is unclear, being almost exclusively documented from diverse DNA sequences commonly found in environmental sequencing datasets from terrestrial and aquatic ecosystems (*Lara et al., 2010*; *Jones et al., 2011*; *Livermore and Mattes, 2013*; *Lazarus and James, 2015*). They appear to be related to but distinct from another lineage, Aphelidea, a group comprised largely of parasites of algae for which genome-scale data is lacking (*Karpov et al., 2014*). Formal descriptions of new species in Rozellomycota are very rare, and the hosts are diverse, ranging from Fungi, slime molds, amoebae, crustaceans, to algae (*Karling, 1942*; *Kagami et al., 2007*; *Corsaro et al., 2014*, *2016*; *Ishida et al., 2015*). These newly described

species possess some characteristics similar to those of *Rozella* and others that more closely resemble Microsporidia.

All Rozellomycota described to date are intracellular parasites that can produce a chitinous cell wall, yet grow as naked protoplasts within their hosts. However, they range greatly in morphology from species that are fungal-like (*Rozella*) and infect using a zoosporic (flagellated) stage to those without flagella where infections occur from spores through a polar filament (*Corsaro et al., 2014*). This latter structure and phylogenomic analyses unite Rozellomycota with the Microsporidia as the earliest diverging lineage of Fungi (*James et al., 2013*; *Haag et al., 2014*). The consideration of Rozellomycota and Microsporidia as members of the Fungi has been contentious due to the fact that they possess some, but not all, of the features characteristic of most Fungi (*Richards et al., 2017*). Some authors consider Rozellomycota to be outside of the Fungi (*Cavalier-Smith, 2013*; *Karpov et al., 2014*), because the genus *Rozella* and Aphelidea perform a form of phagocytosis of host cytoplasm making them appear to lack a defining feature of Fungi, osmoheterotrophy (*Powell et al., 2017*; *Richards et al., 2017*). Others have considered the clade as part of the Fungi, apparently accepting that the features of known species are sufficiently fungus-like (*Hibbett et al., 2007*; *Berbee et al., 2017*; *Spatafora et al., 2017*). Here, we show that all members of this clade are characterized by rampant gene loss, making it likely they are derived from a more fungus-like ancestor.

Microsporidians are represented by approximately 1700 species and are known to infect many animals, including humans (*Keeling, 2009*). They possess a mitochondrial-derived, genome-less organelle called a mitosome (*Hirt et al., 1997*; *Williams et al., 2002*; *Burri et al., 2006*), which has degenerated presumably because they steal ATP directly from their hosts (*Tsaousis et al., 2008*). Their nuclear genomes have also attracted considerable attention for being some of the smallest known for eukaryotes, reaching at the extreme only 2.3 Mb (*Corradi et al., 2010*).

To date, only one genome from the Rozellomycota lineage has been sequenced, that of *Rozella allomycis*, an obligate intracellular parasite of the water mold *Allomyces* (*James et al., 2013*). Phylogenetic analysis of the genome of *R. allomycis* strongly supported the placement of *Rozella* with Microsporidia and revealed previously unsuspected similarities with representatives of the phylum, as it harbored genes previously thought to be Microsporidia-specific. These genes included a nucleotide transporter essential for the acquisition of energy via ATP theft from the host and other genes with obvious signatures of horizontal gene transfers (*James et al., 2013*; *Alexander et al., 2016*). Other biochemical similarities included the presence of a small mitochondrial genome lacking electron transport Complex I and a seemingly degenerate sequence that is extremely AT–rich. The presence of the nucleotide transporter, the degeneracy or loss of the mitochondrion, and the fact that both Microsporidia and *Rozella* associate in the cell with the host's mitochondria (*Hacker et al., 2014*; *Powell et al., 2017*), supported a hypothesis of direct uptake of ATP for nucleotide metabolism and energy from the host to parasite.

The *Daphnia* pathogen, *Mitosporidium daphniae*, is classified as an early diverging Microsporidia; however, it does not reside on the long branch with the rest of Microsporidia and genome drafts revealed that unlike those highly reduced parasites, it possesses a mitochondrial genome that lacks all genes involved in Complex I of oxidative phosphorylation (*Haag et al., 2014*), similar to *R. allomycis*. *M. daphniae* also lacks the ATP transporters that were suggested to facilitate mitochondrial degeneration in *Rozella* and Microsporidia. This pattern suggests that diverse strategies have evolved for parasitism that may be dependent on either host or the environment, as opposed to being lineage-specific. One common theme, however, is that the clade has evolved under repeated loss, going from a species with more genes than the free-living yeasts, for example *Saccharomyces*, which have approximately 5900 genes, to those with the smallest known eukaryotic genomes through the loss of function of genes involved in primary metabolism, presumably because the organism can acquire them from the host. The existence of a degenerate mitochondrial genome in *R. allomycis* and *M. daphniae* supported a model of step-wise degeneration of the respiratory chain in Rozellomycota (possibly following the acquisition of ATP transporters) that could have ultimately resulted in the emergence of genome-less mitosomes in Microsporidia (*Corradi, 2015*).

*Paramicrosporidium saccamoebae* (*Corsaro et al., 2014*) was described as an intranuclear parasite of *Saccamoeba* [Amoebozoa]. Morphologically, it somewhat resembles *M. daphniae* and Microsporidia, as it produces small-sized spores (approximately 1 µm) with a polar filament that lack observable mitochondria. However, ribosomal DNA phylogenies place *P. saccamoebae* within

Rozellomycota, offering an opportunity to examine another member of Rozellomycota and ask questions about the evolution of this clade. Given the relationship between Rozellomycota and Microsporidia, and the fact that both lineages are known only from isolates which are obligate intracellular parasites, we sought to understand whether they are two distinct lineages with unifying changes such as gene loss and horizontal acquisition, or if the clade forms a gradient from fungus-like ancestors to highly reduced endoparasites through a series of nested reductions. Have reductions in gene content occurred many times independently throughout the evolution of these enigmatic, yet ubiquitous clades? Are there common strategies that Rozellomycota and Microsporidia share in drawing energy from their host? To address these questions, we sequenced the genome and transcriptome of *P. saccamoebae*. Our phylogenetic analyses and gene content comparisons revealed that this species is sister to Microsporidia (and not *R. allomycis*), and suggests that Rozellomycota is paraphyletic. Our genome investigations also uncovered a number of *P. saccamoebae* gene losses related to life inside the host nucleus and revealed that it shares more genes with distant lineages of Fungi and presumably therefore to the ancestor of all Fungi than with its closest sequenced relatives. Our study also highlights independent losses and reductions of mitochondrial functions in the evolution of Rozellomycota.

## Results

To acquire the nuclear and mitochondrial genome of *P. saccamoebae*, the microbial community of the *Saccamoeba* host was sequenced and assembled via metagenomic techniques (*Figure 1*). In addition, we sequenced RNA transcripts of *P. saccamoebae* in order to improve genome annotation and to verify expression of genes of interest. Our approaches produced a target nuclear genome assembly of 7.3 Mb in size on 221 scaffolds and possesses 3750 predicted genes (*Table 1*). This size is intermediate between *R. allomycis* (11.8 Mb) and *M. daphniae* (5.6 Mb), while the number of genes is similar to that of *M. daphniae* (3331 genes). The GC content is 47%, which is within the upper range of sequenced Rozellomycota and Microsporidia [34–47%] (*Katinka et al., 2001*; *James et al., 2013*). Analysis of the core eukaryotic genes (CEGs) indicates that the assembly is fairly complete with identification of 89% of complete CEGs and 91% complete or partial CEGs. As an additional measure of assembly continuity, we found that 50% of the genome resides on only 29 scaffolds. Over 76% of the genome is covered by gene space, while repeated content covers only 0.53% of the genome, most of which are simple or low complexity repeats.

Phylogenetic analysis of a concatenated 53 protein data set, with 26,020 amino acid positions in the alignment, reconstructed a monophyletic clade that includes both Rozellomycota and Microsporidia (maximum likelihood bootstrap support [MLBP]=100), as sister to all the rest of Fungi (*Figure 2*). Within the clade, *P. saccamoebae* diverges within Rozellomycota after *R. allomycis* with strong support (MLBP = 100) but diverges before *M. daphniae*, which is still supported as the earliest diverging microsporidian species. With the current nomenclature, Rozellomycota is therefore reconstructed as paraphyletic. Hypothesis testing of alternative placements of *P. saccamoebae* and Microsporidia were all rejected, supporting a paraphyletic Rozellomycota (*Figure 2—figure supplement 1*), and reanalysis of the data following removal of Microsporidia species (excluding *M. daphniae*) also gave support for a monophyletic Kingdom Fungi (*Figure 2—figure supplement 2*), and for the branching order within Rozellomycota observed in analyses with Microsporidia included.

We compared the predicted proteome of *P. saccamoebae* to *R. allomycis*, *M. daphniae*, 10 other Microsporidia, 19 other Fungi spanning the diversity of sequenced fungal lineages, and the outgroup, *Fonticula alba* (*Figure 3A*). We found *P. saccamoebae* shares the most of its 2470 orthogroups, 1975 (80%), with *Spizellomyces punctatus* (*Figure 3B*), a chytrid fungus which has been shown to share a few specific orthogroups with Microsporidia (*Cuomo et al., 2012*). Although a substantial number of *P. saccamoebae* orthogroups are also shared with *R. allomycis* (1,722; 70%) and *M. daphniae* (1,357;55%), it shares more with many other distantly related Fungi, including species in the derived fungal phyla of Ascomycota and Basidiomycota, than with these two closer relatives. The fewest *P. saccamoebae* orthogroups are shared with all other microsporidians sampled, 589 to 664 (24 to 27%). This pattern of orthology is similar to what is seen in both *R. allomycis* (*Figure 3—figure supplement 1*) and *M. daphniae* (*Figure 3—figure supplement 2*), whereby they share a higher percentage of their proteome with distant relatives than with one another. No orthogroups are universally present for Rozellomycota and microsporidians (*Figure 3A*), and very few clusters

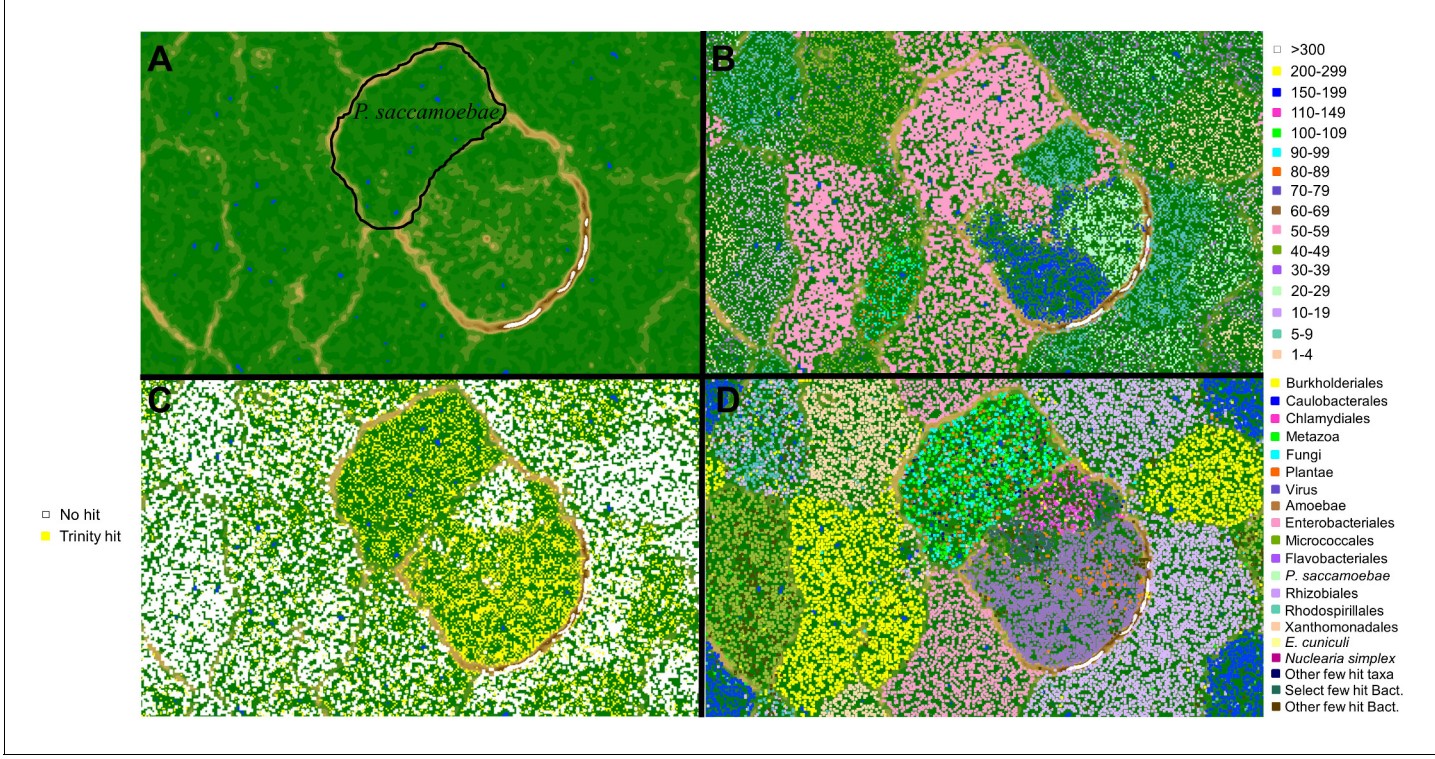

**Figure 1.** Metagenomic plots to decipher target *P.saccamoebae* contigs from metagenomic assembly using multiple lines of evidence. (**A**) Emergent self-organizing map (ESOM) torroidal topology (topo map) with black outline representing the final area surrounding *P. saccamoebae* contigs. Individual points on the map represent 3000 bp windows of metagenomic assembly contigs, whose distance to other points on the map is determined by their tetramer frequency matrix, and green 'valleys' of the map represent windows who frequency matrix are more similar and tan or white 'mountains' separate winodws whose tetramer frequencies are more different. (**B**) Topo map overlaid with contig median coverage; the most common *P. saccamoebae* coverage class (50-59x) is highlighted in pink; this alone was not enough to identify *P. saccamoebae*-specific contigs. (**C**) Contigs with blastn hits to the de novo Trinity assembly are in yellow, and found in many parts of the map including concentrations in the *P. saccamoebae* area, whereas those contigs with no hit to the assembly (in white) are not found in *P. saccamoebae* area. (**D**) Phylogenetic classes mapped onto the topology (based on best blastn to GenBank 'nr' database). *P. saccamoebae* region contains contigs with hits to mostly Fungi, animals, NO HIT, plants, *Nuclearia*, Microsporidia, and the *P. saccamoebae* rDNA. There are a few hits to bacteria that map within the genome, which were further tested to insure they belong on contigs which have multiple fungal proteins on them as well.

DOI: https://doi.org/10.7554/eLife.29594.002

**Table 1.** Genome statistics for *P. saccamoebae* and previously sequenced relatives.

|  | Rozellomycota | | Microsporidia | | | |
|---|---|---|---|---|---|---|
|  | *Rozella allomycis* | *Paramicrosporidium saccamoebae* | *Mitosporidium daphniae* | *Trachipleistophora hominis* | *Encephalitozoon cuniculi* | *Nematocida parisii* |
| Genome size (Mb) | 11.86 | **7.28** | 5.64 | 8.5 | 2.5 | 4.15 |
| GC % | 34.5 | **46.9** | 43 | 34.1 | 47.3 | 34.5 |
| Number of scaffolds | 1059 | **221** | 612 | 310 | 11 | 53 |
| Longest scaffold (bp) | 719,121 | **261,540** | 115,468 | – | – | – |
| L50 | 52 | **29** | 51 | 212 | 6 | 9 |
| # protein models | 6350 | 3750 | 3331 | 3212 | 1996 | 2726 |

DOI: https://doi.org/10.7554/eLife.29594.003

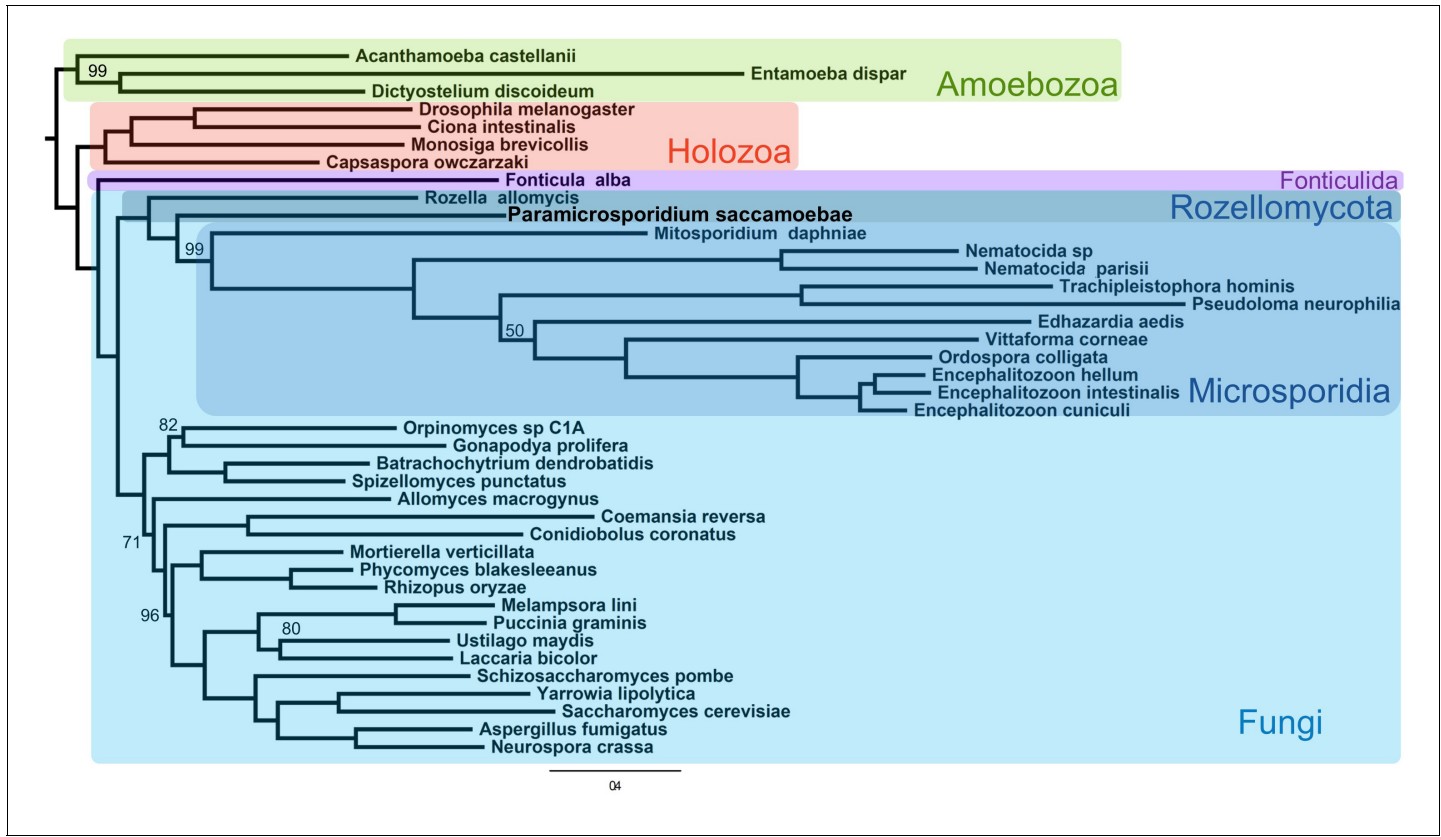

**Figure 2.** Maximum likelihood phylogenomic analysis of 53 proteins with 26,062 amino acid positions in the concatenated alignment and 500 bootstrap replicates. All nodes have 100% bootstrap support except where marked. All Fungi are in shades of blue, and sequenced representatives of Amoebozoa are included as outgroups. Taxa, strains, and references for genomes used are included in *Supplementary file 3*.

DOI: https://doi.org/10.7554/eLife.29594.004

The following figure supplements are available for figure 2:

**Figure supplement 1.** Consel alternative topology likelihood testing results.

DOI: https://doi.org/10.7554/eLife.29594.005

**Figure supplement 2.** Alternative maximum likelihood phylogeny using same data from *Figure 2* and analyzed with same options in RAxML but excluding all Microsporidia except *M.daphniae*.

DOI: https://doi.org/10.7554/eLife.29594.006

were specific to *P. saccamoebae* and either *R. allomycis* (12) or *M. daphniae* (13). There are 377 orthogroups that *P. saccamoebae* shares with other Fungi that are not found in *R. allomycis*, *M. daphniae*, or Microsporidia (*Figure 3A*). Of these, many have no hypothetical annotation, however, eight are related to oxidative phosphorylation functions in the mitochondrion, three are nuclear-pore-associated proteins, and two are proteins involved in converting ethanol to acetate (via alcohol and aldehyde dehydrogenases). Analysis of overall gene loss and gain revealed moderate loss leading to *R. allomycis*, followed by larger amounts of gene loss and lineage-specific gains in *R. allomycis*, *P. saccamoebae*, and *M. daphniae* (*Figure 3C*).

A circular mitochondrial genome that contains a standard complement of fungal mitochondrial genes was assembled onto a single contig of 25,401 bp with 41 predicted genes (*Figure 4*). All genes typically found in fungal mitochondrial genomes that are involved in oxidative phosphorylation are present including those of Complex I (*nad1*, *nad2*, *nad3*, *nad4*, *nad4L*, *nad5*, *and nad6*), Complex III (*cob*), Complex IV (*cox1*, *cox2*, and *cox3*), Complex V (*atp6*, *atp8*, and *atp9*), 23 tRNAs, one ribosomal protein (*rps12*), and the large and small rRNA subunits (*rnL* and *rnS*). This is in contrast to *R. allomycis*, *M. daphniae*, and the derived microsporidians, which have, respectively, lost all Complex I genes or the entire mitochondrial genome altogether (*Figure 4—figure supplement 1*). Like *R. allomycis* and *M. daphniae*, *P. saccamoebae* possesses *rps12*, not *rps3* like most fungal

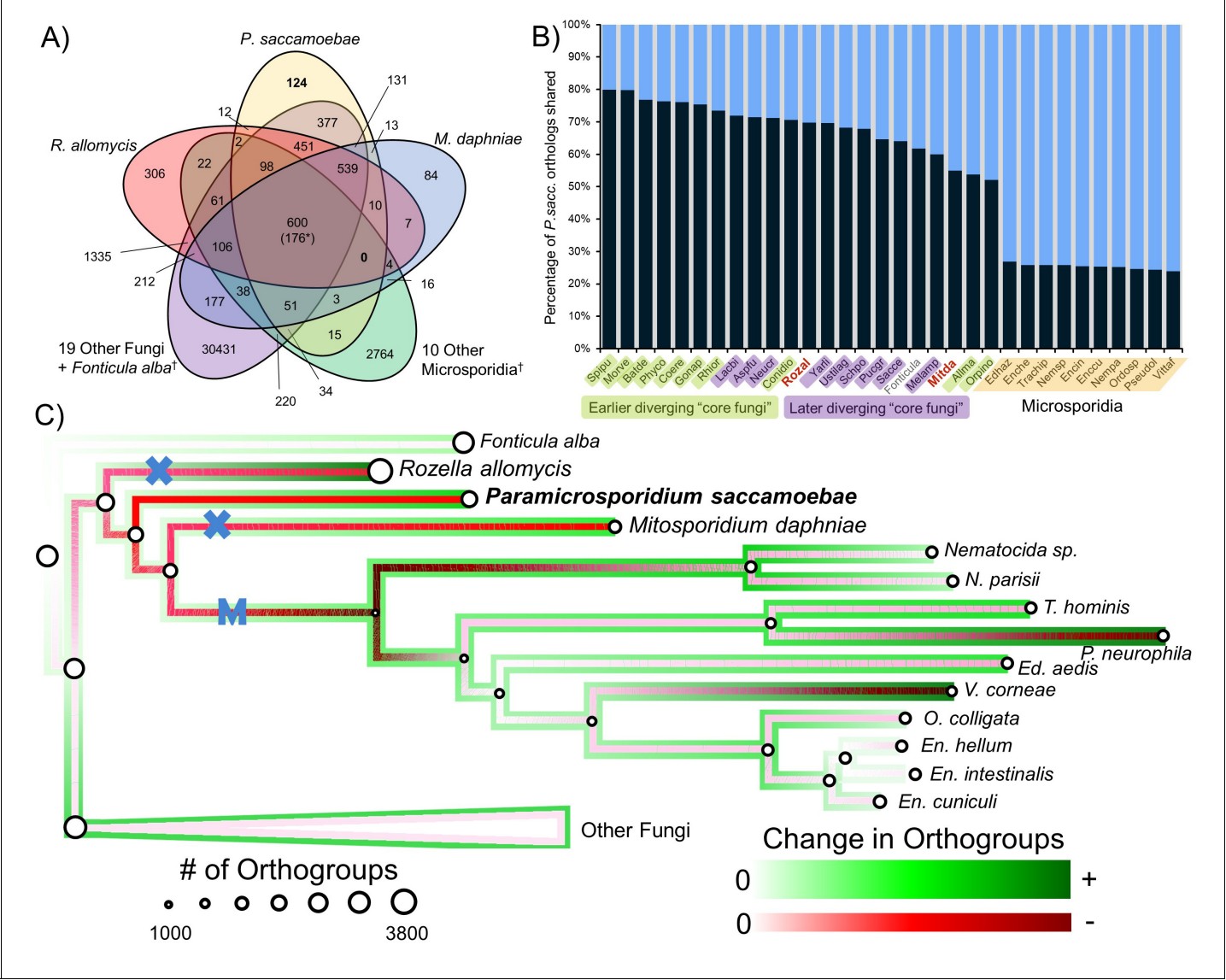

**Figure 3.** Analysis of orthogroups in Rozellomycota and Microsporidia. (A) Venn diagram of orthologous clusters shared and unique to *P. saccamoebae*, related taxa, and other Fungi. *At least one copy present in all 33 taxa. †Numbers reflect presence of at least one species from these groups in a cluster. (B) Graphical representation of percentage of *P. saccamoebae* orthologs shared with other Fungi (and the outgroup *Fonticula alba*). Its closest sequenced relatives, *R. allomycis* and *M. daphniae*, are in red text. (C) Ancestral reconstruction of orthologous cluster gains and losses highlighting the independent reductions in the mitochondrion (X) and loss of true mitochondrion (M) which unites the 'core Microsporidia.' Diameter of circles is relative to the number of orthogroups present in terminal taxa and reconstructed ancestors. To illustrate the quantity of both gains and losses throughout the evolution of this clade, the outside track shows relative amount of orthogroups gained (in shades of green), while the inside track is showing the relative amount of orthogroups lost (shades of red); the lighter the shade of green or red, the fewer respective gains or losses are reconstructed at that node.

DOI: https://doi.org/10.7554/eLife.29594.007

The following source data and figure supplements are available for figure 3:

**Source data 1.** Orthologous cluster raw data for graphs (*Figure 3B*, *Figure 3—figure supplements 1* and *2*) and ancestral reconstruction (*Figure 3C*).
DOI: https://doi.org/10.7554/eLife.29594.010
**Figure supplement 1.** Graphical representation of percentage of *R.allomycis* orthologs shared (darker shade) with other Fungi (and the outgroup *Fonticula alba*).
DOI: https://doi.org/10.7554/eLife.29594.008
**Figure supplement 2.** Graphical representation of percentage of *M.daphniae* orthologs shared (darker shade) with other Fungi (and the outgroup *Fonticula alba*).
DOI: https://doi.org/10.7554/eLife.29594.009

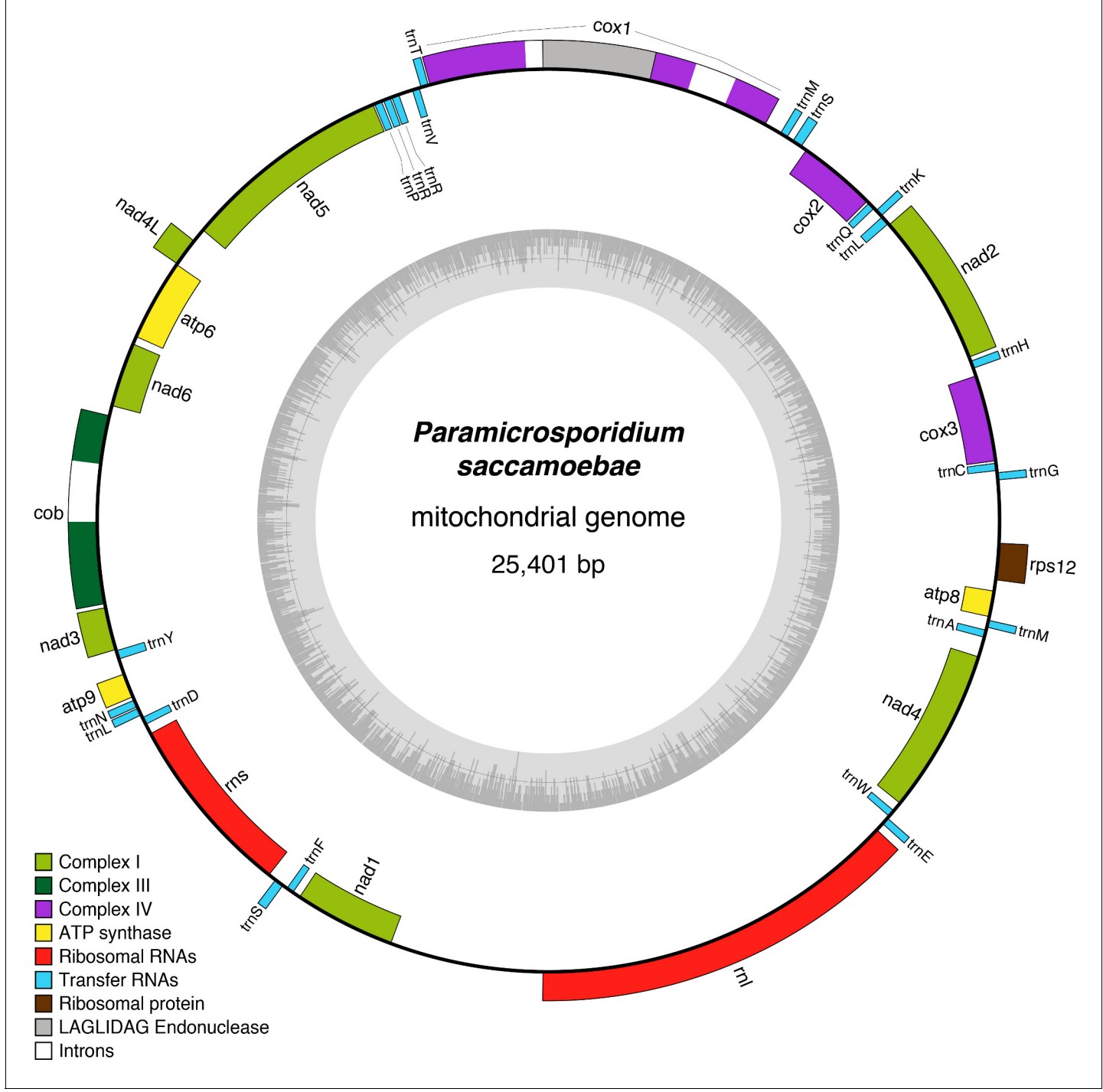

**Figure 4.** Reconstruction of the mitochondrial genome of *P.saccamoebae* with GC Content (represented by height of grey bars - black line is 50%) and gene annotations, which include the seven genes typically found in Fungi involved Complex I of oxidative phosphorylation.

DOI: https://doi.org/10.7554/eLife.29594.011

The following figure supplements are available for figure 4:

**Figure supplement 1.** Presence and absence of mitochondrial genes in *P.saccamoebae* and related species.
DOI: https://doi.org/10.7554/eLife.29594.012

**Figure supplement 2.** Maximum likelihood phylogeny of Mitochondrial genes *cox1*, *cox2*, *cox3*, *cob1*, and *atp6*.
DOI: https://doi.org/10.7554/eLife.29594.013

mitochondrial genomes (**Aguileta et al., 2014**; **Yang et al., 2017**), indicating these proteins have been differentially retained between these two lineages (**Figure 4—figure supplement 1**). Due to the metagenomic nature of this sample, phylogenetic analysis was used to determine if the host (amoebozoan) mitochondrial genome had been inadvertently sequenced. Using the conserved mitochondrial genes (*cob*, *cox1*, *cox2*, *cox3*, and *atp6*), *P. saccamoebae* mitochondrion is placed as sister to *R. allomycis* and *M. daphniae*, and not with the available amoebozoan mitochondrial sequences (**Figure 4—figure supplement 2**). Moreover, supplementary genes involved in the electron transport chain (including ferrodoxin, ubiquinone, succinate dehydrogenase, etc.) are present in the nuclear genome of *P. saccamoebae*, in addition to being expressed, suggesting it possesses full oxidative phosphorylation functionality.

To analyze host dependency and energy acquisition, we analyzed genes involved in nucleotide and amino acid biosynthesis and import. The genes necessary for de novo nucleotide (both purine and pyrimidine) biosynthesis are all absent in *P. saccamoebae*, suggesting it is not capable of producing its own nucleotides, and it also lacks genes for converting nucleosides to nucleotides and vice versa (**Figure 5**). One of the most highly expressed transporter genes is a nucleoside transporter (PSACC_00918), which has homologs in many other Fungi including *M. daphniae*, but which is not found in *R. allomycis*. Another nucleoside transporter (PSACC_02618) was also identified and is expressed. Other highly expressed transporters include an ABC transporter (PSACC_00512), an Na + dicarboxylate and phosphate transporter (PSACC_03652) and a general substrate transporter (PSACC_01530). There is no evidence for the presence of the horizontally acquired thymidine kinases found by **Alexander et al. (2016)** in both *R. allomycis* and several microsporidians.

Many amino acid biosynthesis pathways are reduced or absent in *P. saccamoebae* (**Figure 6A**), and as a result it is likely that it does not produce histidine or tryptophan, and may require intermediates from the host to complete synthesis of the following: lysine, methionine, arginine, phenylalanine, isoleucine, leucine, valine, and proline. For synthesis of branched chain amino acids (valine,

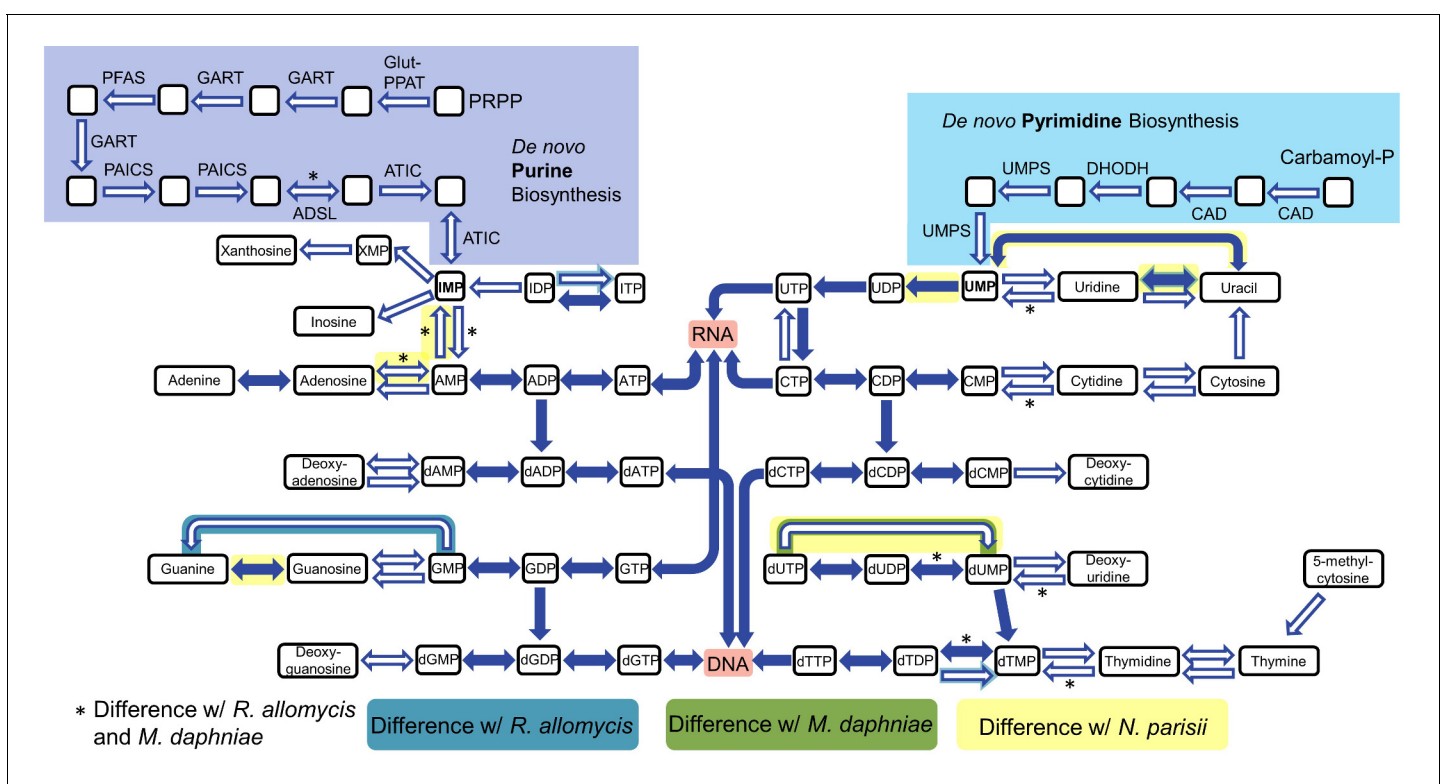

**Figure 5.** Nucleotide biosynthesis pathways present and absent in *P.saccamoebae*, and comparison with related taxa. Hollow arrows are proteins not identified in the *P. saccamoebae* genome. Differences with *P. saccamoebae* are highlighted as specified in the key at bottom. Figure design adapted from **Dean et al. (2016)**.

DOI: https://doi.org/10.7554/eLife.29594.014

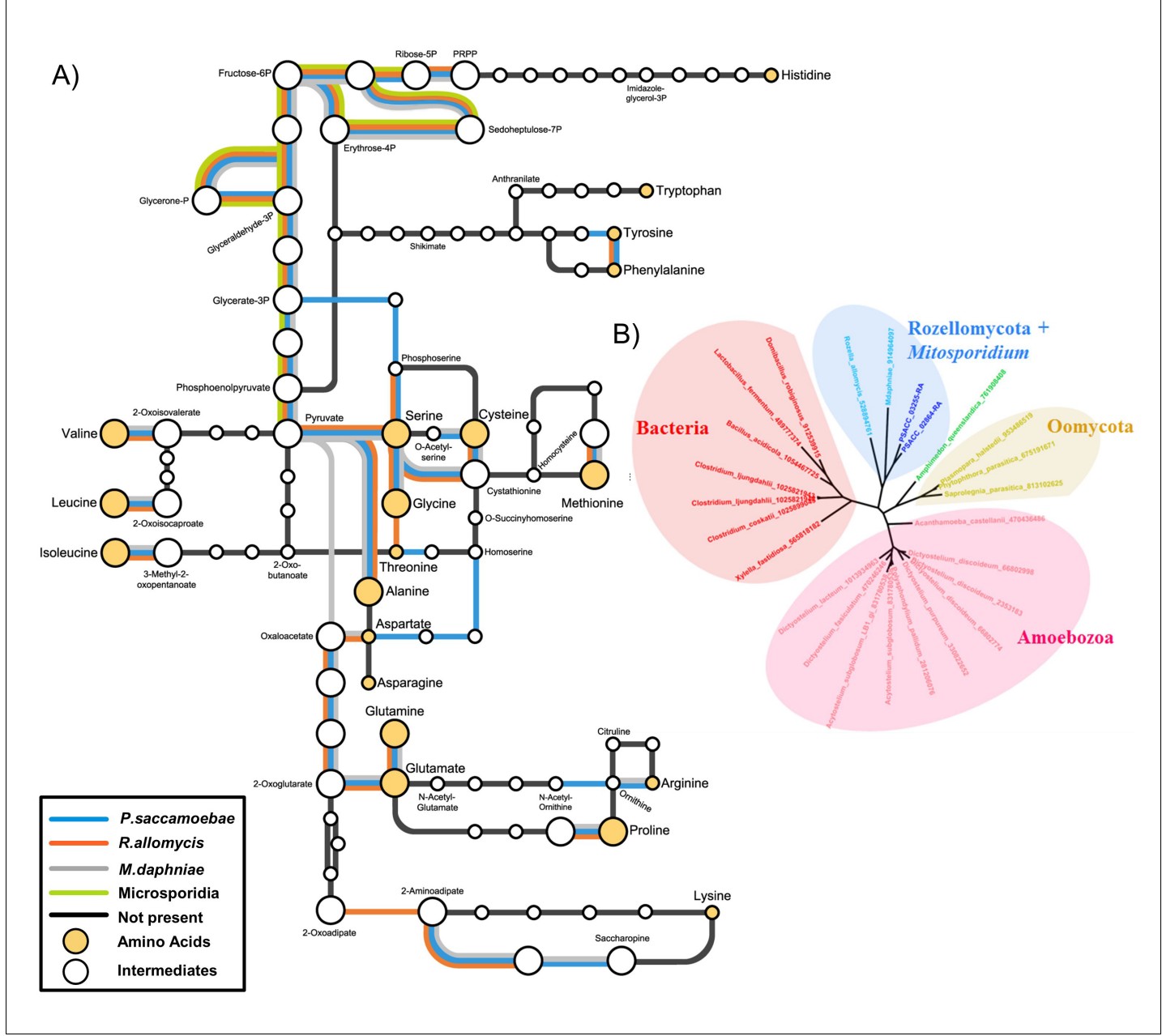

**Figure 6.** Amino acid biosynthesis capability and amino acid permeases in Rozellomycota and *Mitosporidium daphniae* and other Microsporidia. (A) Amino acid biosynthesis map for selected taxa, with end products and intermediates, for *P. saccamoebae*, *R. allomycis*, *M. daphniae*, all of which contain lineage specific enzyme retentions, while other pathways, specifically the ability to produce serine, glycine, alanine, and cysteine are maintained in all three taxa. (B) Phylogenetic analysis of amino acid permeases in *P. saccamoebae*, *R. allomycis*, and *M. daphniae*, which are not found in other Microsporidia or any other Fungi.

DOI: https://doi.org/10.7554/eLife.29594.015

leucine, and isoleucine), a single protein, a homolog of branched chain amino acid aminotransferase (PSACC_01141 in *P. saccamoebae*), is present and expressed based on our mRNA-seq data. Our investigations also revealed similar yet unique profiles of reductions in amino acid biosynthesis enzymes in *R. allomycis* and *M. daphniae* as well (*Figure 6A*), while the rest of Microsporidia cannot produce any amino acids. We identified two expressed amino acid permeases (PSACC_02864 and PSACC_03255) that may offset these gene losses in the *P. saccamoebae* genome, which are homologous to a single permease in both *R. allomycis* and *M. daphniae* and most closely related to

bacterial, not fungal, amino acid permeases (*Figure 6B*). There are three fungal chitin synthase domain containing proteins in *P. saccamoebae* (PSACC_00558, PSACC_00865, PSACC_01336). These are all Division I chitin synthases with homologs in *R. allomycis*. Unlike *R. allomycis*, no Division II chitin synthases were identified, a characteristic *P. saccamoebae* shares with Microsporidia.

P. saccamoebae has almost all of the conventional meiosis-related genes, including all known meiosis-specific genes (*Rec8*, *Spo11-1*, *Dmc1*, *Hop2*, *Mnd1*, *Msh4*, and *Msh5*; *Figure 7A*, *Supplementary file 1*). The only meiosis-related gene absent from the *P. saccamoebae* genome is a homolog of *Mlh3* which is involved in DNA mismatch repair and appears to be absent in all other members of this earliest diverging fungal clade. The number of meiosis genes found in *P. saccamoebae* is greater than any other sequenced Rozellomycota or Microsporidia with available genomes. This indicates that this obligate intracellular parasite is theoretically able to undergo conventional meiotic recombination, which is bolstered by analysis of SNP frequencies that revealed that the vast majority of alleles possess a 50:50 ratio, indicating that *P. saccamoebae* is likely diploid, like most microsporidian species examined to date (*Figure 7B*) (*Cuomo et al., 2012*; *Haag et al., 2013a*, *2013b*; *Selman et al., 2013*; *Ndikumana et al., 2017*; *Pelin et al., 2016*). Other *P. saccamoebae* genes potentially involved in mating include two putative homeodomains (HD1 and HD2 homologs) (PSACC_01945 and PSACC_01946) whose genome organization is strikingly similar to that found in loci that govern sexual identity in more derived Fungi, particularly Basidiomycota. However, heterozygosity and extensive allelic divergence typical of fungal MAT-loci is absent along those genes. Furthermore, their genome organization is not found to be conserved in other Rozellomycota and Microsporidia. Intriguingly, divergent copies of both HD1 and HD2 exist at other locations in the genome of the microsporidians *Pseudoloma neurophilia* and *Nosema ceranae* (*Figure 7—figure*

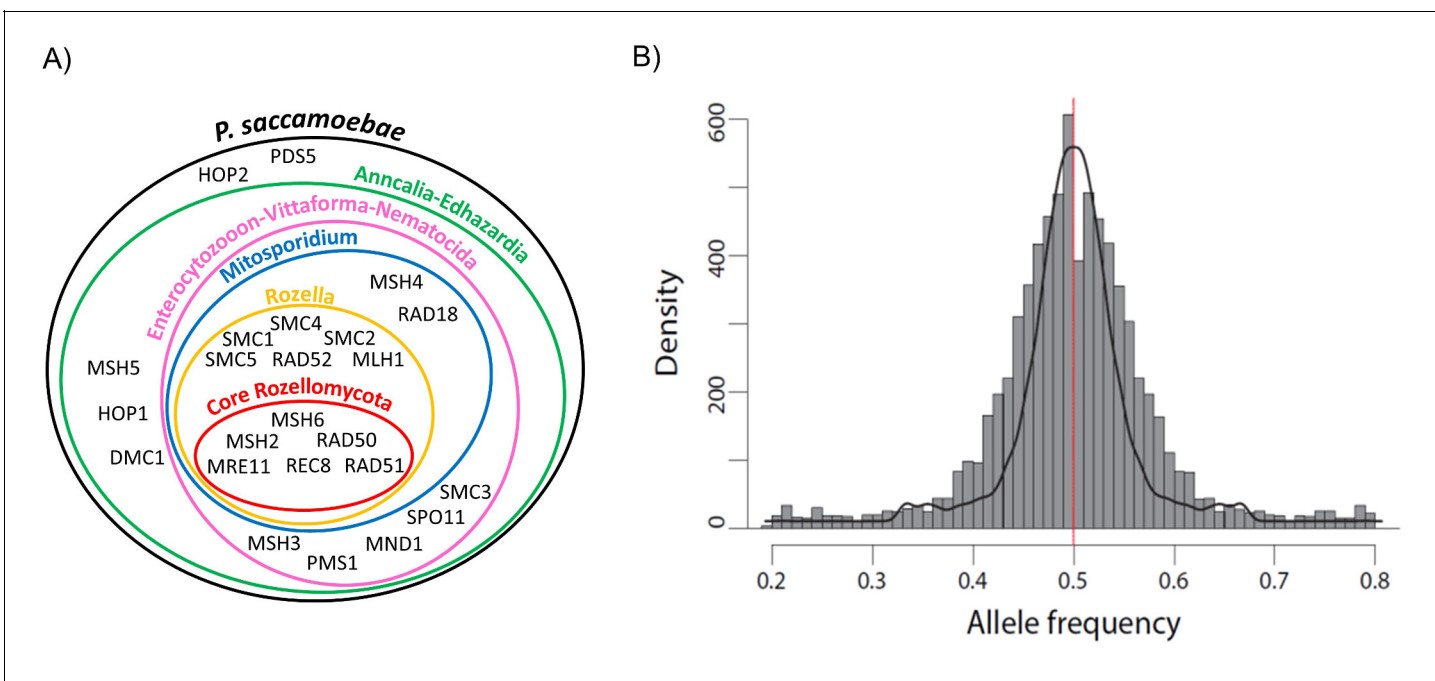

**Figure 7.** Schematic representation of Rozellomycota and Microsporidia meiotic genes and analysis of *P. saccamoebae* allele frequency. (A) Venn diagram showing the presence or absence of the core genes known to be directly involved in meiotic processes. The presence or absence of these genes have been investigated in the genomes of *P. saccamoebae* and its relatives, including representative species belonging to the phylum Microsporidia, described in detail in *Supplementary file 1*. (B) Allele frequency analysis of *P. saccamoebae* genomic scaffolds. The histogram of allele frequency distribution is based on read counts of filtered bi-allelic SNPs and is overlapped by density curves (black). The vertical line (red) represents the 0.5 allelic frequency. A total of 3832 variable nucleotide positions were analyzed.

DOI: https://doi.org/10.7554/eLife.29594.016

The following figure supplement is available for figure 7:

**Figure supplement 1.** Homology and synteny comparison for homeodomain gene clusters in seven Microsporidia and Rozellomycota.

DOI: https://doi.org/10.7554/eLife.29594.017

*supplement 1*), but none of these show the organization or divergence expected for fungal-like MAT loci.

To examine the evolution and history of the flagellum and its losses within Rozellomycota, a search was conducted for the dozens of genes involved in flagellum components and functioning. This revealed that unlike *Rozella* spp. and environmentally acquired Rozellomycota targeted using FISH (*Jones et al., 2011*), *P. saccamoebae* does not possess the ability to produce a flagellum (*Supplementary file 2*). Using hidden Markov models (HMMs) developed using alignments of known polar filament proteins (PFPs) in Microsporidia (*Han and Weiss, 2017*), we searched the predicted proteome of *P. saccamoebae* for sequence-based evidence of PFPs and found no significant hits (all conditional p>0.001). This likely reflects the rapid rate of evolution of these proteins, which are only known from species with a well-developed polar filament. Our HMMs similarly did not detect PFPs from *M. daphniae*, but detected all PFPs in the microsporidian *Encephalitozoon romaleae*.

## Discussion

The existence of a typical fungal mitochondrial genome, with a complete electron transport chain in *P. saccamoebae* is in stark contrast with the hypothesis of gradual reduction of oxidative respiration function within the Rozellomycota-Microsporidia clade eventually leading to the origin of mitosomes in Microsporidia (*Tsaousis et al., 2008*; *James et al., 2013*). Similar to what was found in *M. daphniae*, a mitochondrion was not identified using electron microscopy in the original species description of *P. saccamoebae* (*Corsaro et al., 2014*), indicating that organelles can be difficult to discern in micrographs of these very small (~1 µm) endoparasites. In fact, in both *M. daphniae* (*Haag et al., 2014*), and *P. saccamoebae*, the only evidence for a functional mitochondrion lies in the genomic data, yet interpretation of this data should be considered indisputable. However, the presence of all genes in Complex I of oxidative phosphorylation in *P. saccamoebae*, taken in concert with the phylogenetic results, indicates that the reductions in *R. allomycis* and *M. daphniae* mitochondrial genomes represent independent and convergent losses. Further evidence that *P. saccamoebae* possesses a functional mitochondrion derives from the mitochondrial genome phylogeny (*Figure 4—figure supplement 2*) and the presence and expression of genes for mitochondrial function (e.g. succinate dehydrogenase, NADH-ubiquinone oxidoreductase) in the nuclear genome.

It is unclear why the lifecycle and environment of *P. saccamoebae* has led to maintenance of a mitochondrion capable of all links of the electron transport chain, whereas this ability has been reduced (via loss of the same complex) in the related endoparasites *R. allomycis* and *M. daphniae*. The independent reductions in Complex I suggests that there may have been multiple transitions to intracellular parasitism and that putative 'epiphytic' Rozellomycota are or were once occupying non-parasitic ecological niches (*Jones et al., 2011*; *Ishida et al., 2015*). One explanation for the maintenance of a fully functioning mitochondrion in *P. saccamoebae* could be related to its subcellular location. Studies using light microscopy of *R. allomycis* (*Powell et al., 2017*) and Microsporidia (*Hacker et al., 2014*) have found that these parasites are surrounded by host mitochondria, presumably to facilitate energy theft from the host. Therefore, it could be that *P. saccamoebae*, which is sequestered to the nucleus, is prevented from co-locating with the host mitochondria from which it could steal energy. There are, however, Microsporidia that are intranuclear parasites of fish and crustaceans, for example *Nucleospora* (*Freeman et al., 2013*) and *Enterospora* (*Stentiford et al., 2007*), and they have no mitochondrial genome. *R. allomycis* and many Microsporidia possess horizontally acquired ATP transporters by which they steal ATP from their host, and these were not found in *P. saccamoebae*. Potentially, a lack of these (and the ability to steal energy) has led to the maintenance of Complex I genes in its mitochondrion, although *M. daphniae* (which has lost Complex I) does not possess these ATP transporters as well. Alternatively, the mitochondrion in *P. saccamoebae* could be maintained for use in a perhaps unobserved stage in its lifecycle (although our results have ruled out the possibility of a flagellated stage). This hypothesis is bolstered by the fact that a mitochondrion was not observed in the spores of *P. saccamoebae* via microscopy. In any case, additional genome sequences from this phylum will hopefully clarify how often mitochondrial genomes have been affected by reductive processes (including its complete loss in Microsporidia), and if these reductions have coincided with transitions to parasitism of particular hosts, acquisition of particular transporters, or in specific environmental conditions.

Our results suggest that Rozellomycota is paraphyletic. Microsporidia, which do form a monophyletic clade including the earliest diverging species, *M. daphniae*, are clearly nested within Rozellomycota (*Figure 2*, *Figure 2—figure supplement 1*). Based on the phylogenies published in this and previous (*Haag et al., 2014*) studies, however, *M. daphniae* does not reside on the long branch with the rest of Microsporidia, and there are several cellular (e.g. presence of mitochondrion) and genomic (e.g. ability to produce some amino acids) characteristics, which suggest this species is biologically more similar to Rozellomycota species than with the extremely reduced Microsporidia. Genomic analysis of additional Rozellomycota, such as *Nucleophaga* spp. (*Corsaro et al., 2014*; *2016*) will be vital for understanding whether *M. daphniae* is in a monophyletic group with the Microsporidia or if, as rDNA trees suggest, it is in the Rozellomycota.

Results from the orthologous clustering analysis indicate that *P. saccamoebae* is more similar in gene content to distantly related Fungi, and presumably therefore to the ancestor of all Fungi, than it is to either *R. allomycis* or *M. daphniae* (*Figure 3B*). Shared gene content is clearly not correlated with evolutionary relationships within the Rozellomycota, and there are not obvious genomic synapomorphies that can be used to link Rozellomycota and Microsporidia together. This, in combination with the reconstruction of gains and losses within Rozellomycota (*Figure 3C*) and differential retention and loss of amino acid and nucleotide biosynthesis enzymes (*Figures 5* and *6*) provide evidence that this group of Fungi have been shaped by a series of independent and repeated losses, rather than nested losses toward more derived branches.

The loss of biosynthesis of nucleotides and multiple amino acids in *P. saccamoebae* indicates that like *R. allomycis* and Microsporidia (*Dean et al., 2016*), this species is highly dependent on its host for aspects of both protein synthesis and DNA replication. The highly expressed nucleoside transporter we identified likely facilitates import of nucleosides from the host amoeba, although the absence of any kinases to convert nucleosides to nucleotides is conspicuous. *P. saccamoebae* also lacks a homolog of the nucleotide transporters (identified in *R. allomycis* and Microsporidia) which were horizontally acquired from Chlamydiae (*Tsaousis et al., 2008*), but *M. daphniae* lacks these as well (*Haag et al., 2014*) suggesting these have either been missed in the genome assemblies, that this acquisition happened at least two independent times, or that the gene has been independently lost in both *P. saccamoebae* and *M. daphniae* after horizontal transfer into the ancestor of Rozellomycota. Within this context, the cellular location during parasitism (inside the nucleus) of *P. saccamoebae* may facilitate access to nucleotides from the host, and it may be that the mRNA of the host is phagocytized or translocated into the cell and then broken into essential components by the RNA degradation pathway. The same scenario is perhaps true of amino acids, that is, proteolysis of proteins following phagocytosis or translocation. Such a scenario would indicate an intermediate biology of these organisms which more closely resemble the eukaryotic ancestors from which they evolved, than the rest of the Fungi with chitinous-walled vegetative stages. The amino acid permeases in *P. saccamoebae*, *R. allomycis*, and *M. daphniae*, which are both absent in other Fungi and related to bacterial permeases (*Figure 6B*), suggest that these were present in the common ancestor of Rozellomycota. The permease gene has been duplicated in *P. saccamoebae*, possibly as a result of an increased necessity for amino acids from its amoeba host. In the gene-reduced budding yeast, *Saccharomyces cerevisiae*, amino acid biosynthesis pathways are all complete (*Kanehisa et al., 2014*), which taken in combination with the reduction in these in Rozellomycota and Microsporidia (which are even further reduced) suggests that intracellular parasitism may be driving this trend.

The lack of multiple genes involved in flagellum production in the *P. saccamoebae* genome was not unexpected given the microscopy-based observations of the life cycle, including spores possessing a polar filament. Thus, when or how many times the flagellum has been lost in this group of Fungi remains an open question in Rozellomycota evolution. Clearly, based on in situ hybridization studies and environmental DNA studies, disparate members of the phylum have zoospores with flagella (*Jones et al., 2011*), while others are more Microsporidia-like in their morphology without obvious flagellated stages and Microsporidia-sized spores (*Corsaro et al., 2014*; *2016*). However, this study provides further support for the notion that where a polar filament is present, a flagellum is absent. In *Rozella* spp., the zoospores are the infective agents (*Held, 1973*), and therefore the advent of the infective polar filament may have rendered the flagellum unnecessary. Whether the polar filament has a single origin is uncertain, but likely given its complexity. That searches for PFPs failed likely reflects the rapid rate of sequence evolution (*Slamovits et al., 2004*) and lack of conserved gene synteny rather than an independent origin of the polar filament.

Knowledge of the mode of reproduction of Rozellomycota is essential to fully understand their transmission and life cycle. Sexual reproduction has not been observed in Rozellomycota, while rare microscopy-based evidence for meiosis was reported for two unrelated Microsporidia (*Corradi, 2015*). However, overwhelming genome data points toward the existence of cryptic sexual meiotic recombination in most microsporidians (*Lee et al., 2014*), and we uncovered identical signatures in the genome of *P. saccamoebae*. Although this species harbors many genomic signatures of sexual reproduction (i.e. a complete set of meiosis genes and significant evidence of diploidy, a prerequisite for meiosis), deciphering the reproductive strategies of this obligate intracellular parasite and allied species will likely require population-based genetic approaches. In particular, these seem necessary to expose strain-specific divergent loci potentially involved in compatibility, or reveal evidence of intraspecific gene exchange and recombination.

This high-quality assembly (as evidenced by the CEGMA results, presence of meiosis genes, mitochondrial genome assembly, etc.), especially compared to other sequenced species in the clade, demonstrates the power of sequencing metagenomic samples when pure samples are not readily available. Overall, the genome sequence of *P. saccamoebae* has provided crucial insights into the evolution and biology of this highly diverse but severely undersampled clade. In particular, this study revealed this intranuclear parasite has a mitochondrial genome possessing genes for all complexes involved in oxidative phosphorylation, and has a reduced capacity to produce nucleotides and amino acids, suggesting a dependence on the host for genetic building blocks and protein production but not energy. We also found evidence for independent reductions within the nuclear genomes of Rozellomycota, indicating there may have been multiple transitions to parasitism within the clade and that Microsporidia are a particularly highly derived and reduced lineage. Finally, these results provide further whole genome evidence for the paraphyly of Rozellomycota.

## Materials and methods

### DNA isolation, sequencing, and assembly

Resting spores of *P. saccamoebae* strain KSL3 were obtained from infected *Saccamoeba* sp. grown on bacteria-coated non-nutritive agar plates and cleaned by filtration through a 0.5-μm membrane filter as previously described (*Corsaro et al., 2014*). Whole genomic DNA was extracted using the PowerLyzer PowerSoil DNA Isolation Kit (MOBIO Carlsbad, CA) per manufacturer's instructions amended with an RNase A (Qiagen, Venlo Netherlands) step (6 μL for 30 min incubation at 37° C) and elution volume of 50 μL. A single PacBio SMRTcell was prepared and sequenced at the University of Michigan DNA Sequencing Core (UMDSC). Illumina library construction and DNA sequencing was completed on two lanes of paired end 101 cycle Illumina HiSeq 2000 at UMDSC and resulted in 178,089,240 raw reads. SmrtPortal v2.2.0 was used for assembly of the PacBio data, with the RS_HGAP_Assembly.3 program. This genome assembly with 437 scaffolds, 16.7 Mb, and an average 40x coverage, was exclusively bacterial in nature, and used to remove 'non-target' bacterial raw reads from the Illumina data. Raw Illumina reads were trimmed (of first 5 bases and last 16 bases for a total length of 80 bp) and filtered (all bases q-score $\geq$20) for quality using scripts as a part of the Fastx toolkit (*Gordon, 2011*), and then aligned to the PacBio assembly using Bowtie2 (*Langmead and Salzberg, 2012*) with default settings, resulting in 127,660,572 reads. These remaining reads (those that did not align) were assembled with Velvet v1.2.10 (*Zerbino and Birney, 2008*) with a kmer size of 53 and binned based on coverages estimated with MetaVelvet v1.2.02 (*Namiki et al., 2012*) before subsequent rerun of velvetg that resulted in a final metagenome assembly size of 72 Mb on 3843 contigs. To identify target genome contigs, analysis and visualization of tetramer frequencies was conducted using the emergent self-organizing map (ESOM) program (*Ultsch and Mörchen, 2005*) and associated scripts developed for metagenomic binning (*Dick et al., 2009*; *Anderson et al., 2010*) (*Figure 1*), with default settings and window size of 3000. Visualization within ESOM used the default settings except for the following: K-batch training algorithm, 150 rows, 210 columns, and a start radius value of 50. The resulting subsection of contigs is the one referred to in the results was used for all downstream analyses.

The CEGMA program (v 2.5) was used to find core eukaryotic genes for assessment of genome assembly completeness (*Parra et al., 2007*).

## RNA isolation, sequencing, and gene annotation

Spores of *P. saccamoebae* strain KSL3 previously collected and stored in 70% ethanol were pelleted by centrifugation. The supernatant was discarded and the spores were suspended in 450 μL of RNeasy mini kit (Qiagen) lysis buffer RLC and 200 μl of sterile glass beads (150–212 μM). The sample was incubated for 15 min at 56°C and mechanically disrupted in a shaker at 2500 rpm for 30 s every 5 min. RNA extraction was further performed using the RNeasy mini kit following manufacturer's recommendations and a DNase I digestion on column was done. Final elution was in a volume of 20 μL in buffer TBE. The eluted RNA was quantified on a LabChip GX (PerkinElmer) showing a concentration of 69.3 ng/μL and further processed for library preparation with the Illumina stranded mRNA kit with poly A purification. One full lane of the library was sequenced on a HiSeq 2500 instrument on High-Output V4 paired-reads (2 × 125 bp) mode (Fasteris, Switzerland).

A total of 72,998,800 paired reads were filtered to remove adaptors and quality trimmed (Quality Phred score cutoff: 20) with TrimGalore v0.4.0 (*Krueger, 2015*). The resulting reads were de novo assembled using Trinity v2.1.1 (*Haas et al., 2013*), giving a total of 38,158 transcripts (37,059 predicted trinity genes), a GC content of 42.4%, an N50 of 312 bp. The genome-guided assembly rendered a total of 4612 transcripts (4280 predicted Trinity genes), a GC content of 48.2%, a N50 of 819 bp and a coverage of 1127x, calculated with bedtools (genomecov) v2.17.0 (*Quinlan, 2014*).

The Maker pipeline (*Cantarel et al., 2008*) was used for de novo gene annotation with the default settings except that keep_preds = 1, to insure maximum retention of protein models. The Trinity RNA assembly was used as EST data, and all proteins in the SwissProt reviewed database (accessed May 20, 2016) (*Boeckmann et al., 2003*) plus protein models from early diverging Fungi were used as protein evidence data (*Supplementary file 3*). The Augustus training species was *Rhizopus oryzae* (*Stanke and Morgenstern, 2005*), and a custom repeat library was created using RepeatScout v 1.0.3 (*Price et al., 2005*) and the model organism for RepeatMasker v 4.0.6 was set to 'Fungi' (*Smit et al., 2014*). Whole nuclear and mitochondrial genome sequences and protein annotations have been deposited in GenBank under the accession MTSL00000000.

MFAnnot was used to annotate genes and proteins for the mitochondrial genome (*Beck and Lang, 2010*). Manual annotation of the mitochondrial ribosomal large subunit gene (*rnL*) and NADH dehydrogenase subunit 6 (*nad6*) was conducted via homology searches for conserved regions using blastn procedures against nr and a selected list of LSU sequences from Fungi, and blastp with 'No Adjustments' setting for compositional adjustments against Genbank nr database for *nad6*. OGDRAW was used for visualization of the mitochondrial genome (*Lohse et al., 2007*).

## Functional annotation of gene space

HMMs were built for polar filament proteins (PFPs) based on sequence alignments from Microsporidia species using Hmmer3 hmmbuild function (*Eddy, 2011*). Sequences (*Supplementary file 4*) were initially aligned using MUSCLE v 3.8.31 with default settings (*Edgar, 2004*). We then searched the predicted proteome of *P. saccamoebae* using hmmsearch of Hmmer3. To serve as controls, we applied the HMM searches to the microsporidian species *Encephalitozoon romaleae* and *Mitosporidium daphniae* proteomes.

For analysis of ploidy, the paired-end reads set was mapped against the scaffolds of the best genome assembly using the Burrows–Wheeler alignment tool (BWA) v 0.7.10 (*Li and Durbin, 2009*), with the BWAMEM algorithm. SAMtools v 0.1.19 (*Li et al., 2009*) was used to convert SAM files into sorted BAM. SNPs were called using FreeBayes v 0.9.21 (*Garrison and Marth, 2012*), with the following parameters: -K (i.e. output all alleles that pass input filters), excluding alignments with mapping quality less than 20 (-m 20) and taking into account only SNPs with at least two alternate reads (-C 2). SNPs were filtered to avoid the analysis of false positives (that is, SNPs originating from misalignment and/or paralogs) using the vcffilter tool from the vcf-lib v 1.0.0 (*Garrison, 2012*) library according to (1) the read depth (maximum read depth: DP <1.25 × genome mean coverage; minimum read depth: DP >0.75 × genome mean coverage), (2) the type of SNPs (only considering SNPs, not indels: TYPE = snp), (3) considering only one alternative allele (NUMALT = 1) and (4) the reference allele observation (RO >1). Allelic frequency plots were plotted using the free software environment for statistical computing and graphics, R v3.1.3 (*R Core Development Team, 2013*).

Meiosis-related genes and homeodomain proteins were identified along the genome of *P. saccamoebae* using reciprocal blastx and tblastx procedures using a list of fungal meiosis-related genes

available in *Halary et al. (2011)*. The genomic organization of *P. saccamoebae* HDs was manually compared against those of *R. allomycis* and available Microsporidia available in GenBank.

To identify flagellum-related proteins, a blast database was created using the set of genes identified in *Naegleria fowleri*. Blastp was used to identify *P. saccamoebae* and *M. daphniae* proteins with hits at e-values <1*e-20 (*Supplementary file 2*), and results were compared with those found in *James et al. (2013)*.

For annotation of KEGGs in analyzed taxa, all protein models were uploaded to the KAAS server (*Moriya et al., 2007*), and the 'for Eukaryotes' representative set was chosen and amended with the following taxa: *Neurospora crassa*, *Botrytis cinerea*, *Aspergillus fumigatus*, *Parastagonospora nodorum*, *Tuber melanosporum*, *Ustilago maydis*, *Nosema ceranae*, and *Monosiga brevicollis*. Finally, other functional annotation was performed using BLAST+ v 2.2.29 against the Uniprot TrEMBL database downloaded May 20, 2016 (*Boeckmann et al., 2003*).

## Phylogenomic analyses

To cluster orthologous proteins, the Fastortho implementation (*Wattam et al., 2014*) of orthoMCL (*Li et al., 2003*) was run using default settings and inflation parameter of three with select outgroups and target taxa (*Supplementary file 3*). Ancestral character estimation of orthogroups present or absent in the internal nodes of the tree were estimated using the 'ace' command as a part of the APE package in R (*Paradis et al., 2004*). Data were recoded into a presence/absence binary matrix of each orthogroup, and ancestral states estimated using a symmetrical model (SYM). A 50% inclusion cutoff was used, meaning the orthogroup was reconstructed as being present in the ancestral node if it received greater than 50% marginal likelihood.

For phylogenomic analysis, HMMs were built for each of the 53 protein alignments used in *Capella-Gutiérrez et al. (2012)* and *Haag et al. (2014)* using Hmmer3 (*Eddy, 2011*). HMMs were then used to extract these proteins from the target species (*Supplementary file 3*). Proteins were aligned using MUSCLE v 3.8.31 (*Edgar, 2004*), and gaps were excluded with trimAl (*Capella-Gutiérrez et al., 2009*) with the 'gappyout' setting. All phylogenetic trees of the concatenated alignment were created with Raxml v 8.2.8 (*Stamatakis, 2014*) using the Gamma model of rate heterogeneity and the AUTO option chosen for substitution matrix with 500 bootstrap replicates. Confidence assessment and hypothesis testing was completed using CONSEL (*Shimodaira and Hasegawa, 2001*) to analyze the tree puzzle output of Raxml v8.2.8 with '-f g' and with a newick file with the five topologies in *Figure 2—figure supplement 1*.

# Additional information

## Funding

| Funder | Grant reference number | Author |
| --- | --- | --- |
| National Science Foundation | DEB-1354625 | Timothy Y James |
| Natural Sciences and Engineering Research Council of Canada | | Nicolas Corradi |
| Fonds de Recherche du Québec - Nature et Technologies | | Denis Beaudet |

The funders had no role in study design, data collection and interpretation, or the decision to submit the work for publication.

## Author contributions

C Alisha Quandt, Conceptualization, Data curation, Formal analysis, Investigation, Visualization, Methodology, Writing—original draft, Project administration, Writing—review and editing; Denis Beaudet, Data curation, Formal analysis, Validation, Investigation, Writing—original draft, Writing—review and editing; Daniele Corsaro, Conceptualization, Investigation, Visualization, Methodology, Writing—review and editing; Julia Walochnik, Visualization, Methodology, Writing—review and editing; Rolf Michel, Resources, Methodology, Writing—review and editing; Nicolas Corradi, Conceptualization, Formal analysis, Supervision, Funding acquisition, Validation, Investigation, Writing—

original draft, Writing—review and editing; Timothy Y James, Conceptualization, Resources, Formal analysis, Supervision, Funding acquisition, Validation, Investigation, Methodology, Writing—original draft, Writing—review and editing

### Author ORCIDs
C Alisha Quandt (iD) http://orcid.org/0000-0003-0260-8995
Nicolas Corradi (iD) https://orcid.org/0000-0002-7932-7932
Timothy Y James (iD) http://orcid.org/0000-0002-1123-5986

### Decision letter and Author response
Decision letter https://doi.org/10.7554/eLife.29594.025
Author response https://doi.org/10.7554/eLife.29594.026

## Additional files

### Supplementary files
• Supplementary file 1. Presence (x) and absence (-) of conventional meiosis genes within Microsporidia and Rozellomycota species. Annotations of *P. saccamoebae* meiosis related homologs given in first column.
DOI: https://doi.org/10.7554/eLife.29594.018

• Supplementary file 2. Flagellum-related protein presence or absence in several Fungi, animals, and oomycetes. Across the top row, possession of a flagellum – yes, no, or unknown (?). Amended to and modified from *James et al. (2013)*.
DOI: https://doi.org/10.7554/eLife.29594.019

• Supplementary file 3. Information and references about genome data used for all comparative analyses (*Figure 3*) and for protein annotation in the Maker pipeline.
DOI: https://doi.org/10.7554/eLife.29594.020

• Supplementary file 4. Polar filament proteins (PFPs) and their accession numbers used to search *P. saccamoebae* genome.
DOI: https://doi.org/10.7554/eLife.29594.021

• Transparent reporting form
DOI: https://doi.org/10.7554/eLife.29594.022

### Major datasets
The following dataset was generated:

| Author(s) | Year | Dataset title | Dataset URL | Database, license, and accessibility information |
|---|---|---|---|---|
| Quandt CA | 2015 | KSL3_genome_submission | https://www.ncbi.nlm.nih.gov/protein/MTSL0000000 | Publicly available at the NCBI GenBank (accession no. MTSL0000000) |

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
