## [Decision Letter]

Thank you for submitting your article "The genome of an intranuclear parasite unveils alternative adaptations to obligate intracellular parasitism" for consideration by *eLife*. Your article has been favorably evaluated by Diethard Tautz (Senior Editor) and three reviewers, one of whom is a member of our Board of Reviewing Editors. The following individual involved in review of your submission has agreed to reveal his identity: Jeremy Wideman (Reviewer #2).

The reviewers have discussed the reviews with one another and the Reviewing Editor has drafted this decision to help you prepare a revised submission.

Summary:

The study by Quandt and collaborators describes the genome of the intracellular parasite *Paramicrosporidium saccamoebae* in the early fungal lineage Rozellomycota. The authors show that the *P. saccamoebae* genome is not as reduced (in terms of gene content) as the genomes of previously sequenced Rozellomycota or of previously sequenced Microsporidia and also contains a full mitochondrial genome (again in contrast to previously sequenced genomes from this part of the tree of life). Combined with the placement of *P. saccamoebae* on the fungal phylogeny (it appears to have diverged after the divergence of Rozella, another Rozellomycota genome), analysis of the gene content of this species suggests that the path toward adaptation to a intracellular lifestyle was not linear, but likely involved multiple rounds of gene losses and perhaps multiple transitions to parasitism.

Essential revisions:

1) Many microsporidia localize to the nucleus (e.g. Nucleospora and Enterospora see: PMC3606367 and pubmed id 17523545). This means that mitochondrial function does not correlate with nuclear localization. Therefore one of the major hypotheses that the authors put forth, namely that cellular localization correlates with retention of mitochondrial function is refuted. Furthermore, some misconceptions about mtDNAs seem to be held by the authors. Firstly, a "complete and fully functional mitochondrial genome" is a meaningless statement as many mitochondrial genomes contain very few genes but are fully functional (for example, dinoflagellates, and many apicomplexa encode very few mitochondrial proteins but still have aerobic fully functional mitochondria). Also, many mitochondrial genomes across the eukaryotes lack Complex I (including *S. cerevisiae*) and numerous other aerobic/facultative species (e.g. some algae like *Chlamydomonas* see Kamikawa et al. (PMID 26833505). Thus, convergent simplification of the mitochondrial genome is extremely common, and is in fact the major mode of mitochondrial evolution. However, this does not mean that there is not an interesting mitochondrial story in these data. Specifically, even though the reduction of genes may not be that interesting, it may point towards a different hypothesis, namely that "the cryptomycotan ancestor was a parasite with alternating lifecycle stages between an aerobic flagellate and an anaerobic intracellular non-flagellate". If mitochondria (with intact cristae) are not easily visible in the lifecycle stages that are being viewed, perhaps the mitochondrion is not functioning aerobically at this time. This would support a hypothesis that the intracellular phase is anaerobic. Perhaps the motile stage for Cryptomycota like *Rozella*, or infection/germination stage for Cryptomycota like *Paramicrosporidium*, requires oxidative phosphorylation for some reason. This would be a very attractive model for how ETC components (or the flagellum) could be readily (and convergently) lost. The authors are advised to rewrite the mitochondrial section(s) in a way that provides more information on comparisons between other holomycotan taxa and a reformulation (or omission) of the correlative hypothesis.

2) The authors consider the Rozellomycota belongs to the Fungi. This is not correct as 1) they are phagotrophic (not osmotrophic), 2) they do not have the chitinous cell wall at the vegetative stage. Otherwise, if they are included in the Fungi the diagnosis of the latter has to be cardinally changed, and in this case the other holomycots (amoebae *Nuclearia, Parvularia, Fonticula*) must be included in the Fungi. The authors are advised to discuss this point in the Introduction.

3) The overall presentation of the manuscript was below par; many of the figures were unclear, several terms were incorrectly used, and the manuscript contained several typos –. The authors are advised to carefully examine and address all of them and to carefully proof the revised manuscript for both science and style before resubmission.

[Editors' note: further revisions were requested prior to acceptance, as described below.]

Thank you for resubmitting your work entitled "The genome of an intranuclear parasite, *Paramicrosporidium saccamoebae*, reveals alternative adaptations to obligate intracellular parasitism" for further consideration at *eLife*. Your article has been favorably evaluated by Diethard Tautz (Senior Editor) and three reviewers, one of whom is a member of our Board of Reviewing Editors.

The manuscript has been improved but there are a few remaining issues raised by reviewer #2 that i would like you to consider before acceptance:

*Reviewer #1:*

The authors have satisfactorily addressed my comments.

*Reviewer #2:*

In general I am happy with the changes in the revision. This was a pleasurable review process.

The only thing that I think must change is the claim that fungi evolved from an amoebozoan ancestor.

e.g. the fifth paragraph of the Discussion is troublingly inaccurate. Fungi did not evolve from an amoebozoan ancestor… Amoebozoans form a very different group. *Fonticula* and *nuclearia* are not amoebozoans (they are technically amoeba, but this is an extremely derived state – almost as derived as fungi or microsporidia). *Fonticula* and *Nuclearia* are *sister* to fungi which means they shared a common ancestor, but the common ancestor of these two groups is likely to be a simple phagotrophic flagellate (just like the ancestor to *every* major eukaryotic group) perhaps with an amoeboid stage, but perhaps not.

*Reviewer #3:*

The manuscript has been seriously rewritten and looks now more complete. Concerning my worries about the incorrect inclusion of opisthosporidians in the Fungi: the reply of the authors doesn't suit me well. At the same time, the addition in the manuscript Introduction is correct. I understand that this question is still discussable and certainly does not affect the matter of the paper – I'd accept this part of manuscript as ready for publication.

---

## [Author Response]

Essential revisions:1) Many microsporidia localize to the nucleus (e.g. Nucleospora and Enterospora see: PMC3606367 and pubmed id 17523545). This means that mitochondrial function does not correlate with nuclear localization. Therefore one of the major hypotheses that the authors put forth, namely that cellular localization correlates with retention of mitochondrial function is refuted. Furthermore, some misconceptions about mtDNAs seem to be held by the authors. Firstly, a "complete and fully functional mitochondrial genome" is a meaningless statement as many mitochondrial genomes contain very few genes but are fully functional (for example, dinoflagellates, and many apicomplexa encode very few mitochondrial proteins but still have aerobic fully functional mitochondria). Also, many mitochondrial genomes across the eukaryotes lack Complex I (including S. cerevisiae) and numerous other aerobic/facultative species (e.g. some algae like Chlamydomonas see Kamikawa et al. (PMID 26833505). Thus, convergent simplification of the mitochondrial genome is extremely common, and is in fact the major mode of mitochondrial evolution. However, this does not mean that there is not an interesting mitochondrial story in these data. Specifically, even though the reduction of genes may not be that interesting, it may point towards a different hypothesis, namely that "the cryptomycotan ancestor was a parasite with alternating lifecycle stages between an aerobic flagellate and an anaerobic intracellular non-flagellate". If mitochondria (with intact cristae) are not easily visible in the lifecycle stages that are being viewed, perhaps the mitochondrion is not functioning aerobically at this time. This would support a hypothesis that the intracellular phase is anaerobic. Perhaps the motile stage for Cryptomycota like Rozella, or infection/germination stage for Cryptomycota like Paramicrosporidium, requires oxidative phosphorylation for some reason. This would be a very attractive model for how ETC components (or the flagellum) could be readily (and convergently) lost. The authors are advised to rewrite the mitochondrial section(s) in a way that provides more information on comparisons between other holomycotan taxa and a reformulation (or omission) of the correlative hypothesis.

We thank the reviewers for raising these interesting points. We agree with a number of them, while holding slightly different opinions about others. Firstly, our statement of "complete and fully functional mitochondrial genome" is indeed an inaccurate characterization, and the manuscript has now been revised to address this point.

The loss of genes involved in Complex I leads a reduction in the primary mitochondrial function of generating energy in the form of ATP through oxidative phosphorylation. The point that this loss is common throughout eukaryotes and clearly not a barrier to aerobic growth in these organisms is well taken, and a point we observed when sequencing the first Rozellomycota genome (James et al. 2013). However, we developed the hypothesis, or rather expanded on the existing hypothesis (Tsaousis et al. 2008), that mitochondrial loss of function was related to acquisition of ATP transporters from Chlamydia by horizontal transfer. We have recently demonstrated that *Rozella* and Microsporidia both share the traits of genomes encoding ATP transporters and intimate association of host mitochondrial with the parasite membrane (Powell et al. 2017). This led to the hypothesis which we presented in this paper that the separation of *Paramicrosporidium* from the mitochondria by the host nuclear membrane would lower access to host ATP generated via oxidative phosphorylation and lead to differences in numerous aspects of energy metabolism. Indeed, the primary result of the mitochondrial data we obtained for *Paramicrosporidium* indeed does not show this loss of OXPHOS function, and that is the point that we are trying to make about the mitochondrion. Prior to our study, both *Rozella* and *Mitosporidium* were found to have mitochondrial genomes that were both reduced (compared to typical fungal mito genome and with *Nuclearia*) in that they lack all of Complex I of oxidative phosphorylation, and our hypothesis was that this loss corresponded to an ancient loss in the early evolution of this clade and potentially an ancient switch to intracellular parasitism. However, that is not what we found. We found all genes typically found in fungal oxidative phosphorylation, which suggests independent losses in both *Rozella* and *Mitosporidium*, and potentially suggests there have multiple transitions to a parasitic lifestyle within the clade. To further explain this gene composition to the reader, we have added Supplementary file 2 where we show genes present in target taxa, other exemplar fungi, and *Nuclearia*. Flagella use a lot of ATP, and yet *Rozella* spp. (the only Rozellomycota species possessing flagella) lack Complex I. In the paper, we clearly note that we searched for genes typically present in species with a flagellum, and *Paramicrosporidium* is missing these, providing further support that we have not “missed” a flagellated stage in its lifecycle. Several co-authors (including the species original descriptors) have spent a considerable amount of time looking at these organisms. We are aware that some Microsporidia are intranuclear parasites. However, we disagree that our hypothesis for the biology and evolution of *Paramicrosporidium* is necessarily invalidated by this. Specifically, the ancestor to “core Microsporidia” (those on the extremely long branch) lost their mitochondrial genomes in what appears to be a lineage-wide event. This would suggest that the ancestors to *Paramicrosporidium* (which had a mitochondrion capable of oxid. Phos.) and *Nucleospora* and *Enterospora* (which did *not* have a mitochondrion capable of oxid. Phos, but may have had ATP transporters by which to steal ATP) have evolved to exist in similar habitats with very different original molecular “toolkits”. Thus, there is no reason to say it is impossible that these parasites in the same niche cannot have evolved different mechanisms by which to tolerate the inter-nuclear environment.

We have removed the language in the Abstract hypothesizing that sub-cellular location of the parasite may have increased the selection pressure for a functioning mitochondrion. We have left this hypothesis in the Discussion, but have also added a caveat about intranuclear microsporidians *Enterospora* and *Nucleospora* along with a note that these species (and *R. allomycis*) likely have transporters by which to steal ATP from their hosts, while *P. saccamoebae* does not. We have also added an alternative hypothesis as suggested by the reviewer that the mitochondrion could function in some other unseen part of the *P. saccamoebae* lifecycle – although not a flagellated stage, which our results refute.

2) The authors consider the Rozellomycota belongs to the Fungi. This is not correct as 1) they are phagotrophic (not osmotrophic), 2) they do not have the chitinous cell wall at the vegetative stage. Otherwise, if they are included in the Fungi the diagnosis of the latter has to be cardinally changed, and in this case the other holomycots (amoebae Nuclearia, Parvularia, Fonticula) must be included in the Fungi. The authors are advised to discuss this point in the Introduction.

We thank the reviewers for raising these interesting points. This is an active area of current debate, namely whether or not to include the Rozellomycota/Microsporidia in the Fungi or to consider them as sister groups. Presently, there is no simple resolution as this is a debate fueled by a taxonomic passion of certain authors, including many on the author list (James and Berbee 2012; Karpov et al. 2014). On the whole, the mycological community has accepted that both Microsporidia and Rozella/Cryptomycota may be included in the Fungi (NCBI, Stajich et al.; Berbee et al. 2017) and this can also be witnessed in the myriad of fungal ecology studies which are uncovering Rozellomycota at a fast pace. Ironically, the biological community has readily accepted the Microsporidia as Fungi (Adl et al. 2012), but the finding of phagotrophy-like nutrition in *Rozella* has been an impediment to their full acceptance as Fungi. The discovery of Cryptomycota/Rozellomycota created a controversy through the discovery of taxa that seemed somewhat like Fungi and somewhat like protistan ancestors because they possessed symplesiomorphic traits not observed in other fungi (flagellation, possible phagotrophic nutrition, and lack of cell wall) (Jones et al. 2011). Further analysis shows that ultimately *Rozella* has a number of traits that align them with fungi, such as chitinous cell wall in both spore and cyst phase, polarized cell growth using myosin linked chitin synthases (which likely undergo growth through osmoheterotrophy), and development of chytrid like spores in a zoosporangium. The results we report here show that loss was widespread and radically changed the genomes of all extant Rozellomycota. Very likely the common ancestor to Fungi+Rozellomycota+Microsporidia possessed a set of traits that were lost. Ultimately if the definition of Fungi needs to be slightly amended so be it; there is no synapomorphy for the Fungi (Richards et al. 2017). If we can accept that corals with zooxanthellae are still animals, then we can accommodate Rozellomycota into Fungi. After all, the majority of members in the lineage do not undergo any form of phagocytosis, and even *Rozella* phagocytosis is very distinct from that of nucleariids phagocytosing whole cells with cell walls and digesting them.

To address these points, we have added a short discussion of this to the Introduction. Our intention here, though, is not one of where to draw nomenclatural lines outside of the Rozellomycota+Microsporidia clade. Our main systematics point is that more and more evidence (including ours presented) is mounting that Microsporidia are nested within Rozellomycota (Cryptomycota), rendering Rozellomycota a paraphyletic group. We contend that most of the mycological community (to which most of the authors belong) refer to both *Rozella*, Rozellomycota, and Microsporidia as fungi (James and Berbee 2012; Han and Weiss 2017; Spatafora et al. 2017; Stajich 2017), and therefore we refer to them as such in the paper. Finally, we have recently suggested that the simplest definition of the Fungi is the sister group to the Holozoa (Berbee et al. 2017). A definition that requires fungi to possess osmotrophic nutrition could exclude some early-diverging taxa in the fungal clade while failing to exclude osmotrophic members of the Holozoa. Ultimately the higher classification arguments are interesting fodder for discussion, but that lies beyond the scope of this manuscript, and more data on Aphelids are needed before we know how they are related to the other groups (Karpov et al. 2014).

3) The overall presentation of the manuscript was below par; many of the figures were unclear, several terms were incorrectly used, and the manuscript contained several typos. The authors are advised to carefully examine and address all of them and to carefully proof the revised manuscript for both science and style before resubmission.

We have made changes to figures, legends, and the text where appropriate to improve readability.

[Editors' note: further revisions were requested prior to acceptance, as described below.]

Reviewer #2:In general I am happy with the changes in the revision. This was a pleasurable review process.The only thing that I think must change is the claim that fungi evolved from an amoebozoan ancestor.e.g. the fifth paragraph of the Discussion is troublingly inaccurate. Fungi did not evolve from an amoebozoan ancestor… Amoebozoans form a very different group. Fonticula and nuclearia are not amoebozoans (they are technically amoeba, but this is an extremely derived state – almost as derived as fungi or microsporidia). Fonticula and Nuclearia are sister to fungi which means they shared a common ancestor, but the common ancestor of these two groups is likely to be a simple phagotrophic flagellate (just like the ancestor to every major eukaryotic group) perhaps with an amoeboid stage, but perhaps not.

Reviewer #2 made the following comment: “The only thing that I think must change is the claim that fungi evolved from an amoebozoan ancestor.” We have made changes in the text to reflect this, and sincerely regret the choice of wording here. The text now reads, “… the eukaryotic ancestors from which they evolved[…]”.